# One Year of Outpatient Dialectical Behavioral Therapy and Its Impact on Neuronal Correlates of Attachment Representation in Patients with Borderline Personality Disorder Using a Personalized fMRI Task

**DOI:** 10.3390/brainsci13071001

**Published:** 2023-06-28

**Authors:** Ariane Flechsig, Dorothee Bernheim, Anna Buchheim, Martin Domin, Renate Mentel, Martin Lotze

**Affiliations:** 1Functional Imaging Unit, Department of Diagnostic Radiology and Neuroradiology, University of Greifswald, 17475 Greifswald, Germany; ariane.flechsig@stud.uni-greifswald.de (A.F.); martin.domin@uni-greifswald.de (M.D.); 2Department of Psychiatry and Psychotherapy, University Hospital of Greifswald, 17475 Greifswald, Germany; dorothee.bernheim@uniklinik-ulm.de (D.B.); renate.mentel@med.uni-greifswald.de (R.M.); 3Department of Child and Adolescent Psychiatry and Psychotherapy, University Hospital of Ulm, 89075 Ulm, Germany; 4Department of Psychology, University of Innsbruck, 6020 Innsbruck, Austria; anna.buchheim@uibk.ac.at

**Keywords:** neuronal correlates of attachment, borderline personality disorder, adult attachment projective picture system, dialectical behavioral therapy, fear of abandonment and aloneness, social pain, anterior medial cingulate cortex, amygdala, superior temporal sulcus

## Abstract

(1) Background: BPD is characterized by affect dysregulation, interpersonal problems, and disturbances in attachment, but neuroimaging studies investigating attachment representations in BPD are rare. No study has examined longitudinal neural changes associated with interventions targeting these impairments. (2) Methods: We aimed to address this gap by performing a longitudinal neuroimaging study on n = 26 patients with BPD treated with Dialectic Behavioral Therapy (DBT) and n = 26 matched healthy controls (HCs; post intervention point: n = 18 BPD and n = 23 HCs). For functional imaging, we applied an attachment paradigm presenting attachment related scenes represented in drawings paired with related neutral or personalized sentences from one’s own attachment narratives. In a prior cross-sectional investigation, we identified increased fMRI-activation in the human attachment network, in areas related to fear response and the conflict monitoring network in BPD patients. These were especially evident for scenes from the context of loneliness (monadic pictures paired with individual narrative sentences). Here, we tested whether these correlates of attachment representation show a near-to-normal development over one year of DBT intervention. In addition, we were interested in possible associations between fMRI-activation in these regions-of-interest (ROI) and clinical scores. (3) Results: Patients improved clinically, showing decreased symptoms of borderline personality organization (BPI) and increased self-directedness (Temperament and Character Inventory, TCI) over treatment. fMRI-activation was increased in the anterior medial cingulate cortex (aMCC) and left amygdala in BPD patients at baseline which was absent after intervention. When investigating associations between scores (BPI, TCI) and functional activation, we found significant effects in the bilateral amygdala. In contrast, aMCC activation at baseline was negatively associated with treatment outcome, indicating less effective treatment effects for those with higher aMCC activation at baseline. (4) Conclusions: Monadic attachment scenes with personalized sentences presented in an fMRI setup are capable of identifying increased activation magnitude in BPD. After successful DBT treatment, these increased activations tend to normalize which could be interpreted as signs of a better capability to regulate intensive emotions in the context of “social pain” towards a more organized/secure attachment representation. Amygdala activation, however, indicates high correlations with pre-treatment scores; activation in the aMCC is predictive for treatment gain. Functional activation of the amygdala and the aMCC as a response to attachment scenes representing loneness at baseline might be relevant influencing factors for DBT-intervention outcomes.

## 1. Introduction

### 1.1. Characterization of BPD

Borderline personality disorder is the most common personality disorder in psychiatric context [1,2], occurring equally in women and men with typical initial manifestation in adolescence [3,4]. The disorder is associated with considerable functional impairment, intensive treatment utilization, and high societal costs. Psychotherapy is the treatment of choice for BPD. The risk of self-mutilation and suicide is high. In the general adult population, the lifetime prevalence of BPD has been reported to be from 0.7 to 2.7%, while its prevalence is about 10% in outpatient and 20% in inpatient psychiatric services. Patients diagnosed with BPD are characterized by instability of self-image, interpersonal relationships, and affects. In detail, they show unstable interpersonal relationships, interpersonal hypersensitivity, avoidance of abandonment, cognitive self-disturbance, affective instability, and behavioral dysregulation evident in self-harming behavior or suicidal ideation [2,5,6]. Further symptoms include impulsivity, intense anger, feelings of emptiness, strong abandonment fears, suicidal or self-mutilation behavior, and transient stress-related paranoid ideation or severe dissociative symptoms. There is convincing evidence to suggest that the interaction between genetic factors and adverse childhood experiences plays a central role in the etiology of BPD. Prospective longitudinal studies observed impaired caregiving in combination with parental depression or Posttraumatic Stress Disorder (PTSD) and physical abuse and neglect as risk factors for BPD development [7,8]. 

### 1.2. Studies on Attachment

Attachment researchers have convergently found a strong association between borderline pathology, adverse childhood experiences, and insecure, disorganized/unresolved internal working models of attachment. Dysregulated attachment as well as intolerance of being alone are considered to have their roots in insecure relationships with the caregiver, characterized by abuse, neglect, or abandonment [9]. In attachment research, adult attachment can be assessed with reliable and valid measures. One of them is the Adult Attachment Projective Picture System (AAP) validated on the Adult Attachment Interview (AAI) [10]. The AAP classifies the well-established secure, dismissing, preoccupied, and unresolved attachment categories on the basis of verbatim narratives to seven attachment related AAP scenes [11,12]. Various studies found the predominance of preoccupied and especially unresolved attachment in BPD [13,14,15,16,17,18,19]. In addition, studies reported a high number of “traumatic fear indicators” (e.g., words such as “helplessness”, “emptiness” or “suicide”) in the participants’ AAP narratives, particularly in stories to pictures representing aloneness, called “monadic AAP scenes” [20,21]. Unresolved attachment is defined as a breakdown of organized forms of affective regulation that is conceived as the dysregulation of the attachment system [22,23]. When confronted with an attachment-relevant stimulus such as being abandoned or left alone, the internal attachment system is activated on neurobiological, behavioral, and representational levels [21,24,25]. 

### 1.3. fMRI: Paradigms and Findings

Abnormalities in the amygdala, anterior cingulate cortex (ACC), and hippocampus in the neurobiology of BPD have been consistently implicated in comprehensive reviews [26]. More specifically, neural processes linked to human attachment and social exclusion have been repeatedly studied in fMRI studies. The anterior midcingulate cortex (aMCC) neural network associated with processing attachment relevant stimuli as social exclusion has repeatedly been the subject of functional imaging studies on HCs, mostly by using “Cyber-Ball” as a common fMRI-paradigm causing “social pain” [22,27,28]. Using attachment style questionnaires in fMRI studies, two study groups reported a relationship between attachment anxiety and heightened neural activity in aMCC [29,30]. 

The first investigation, using an fMRI-adjusted AAP paradigm in healthy controls with unresolved attachment, found increased activity in the right inferior frontal cortex, the left superior temporal sulcus (STS), the left head of the caudate nucleus, the bilateral temporal lobe, the left amygdala-hippocampus area, as well as the right amygdala [31]. In a following study, BPD patients were compared with HCs using the same paradigm. The results showed that only the patients with unresolved (vs. organized) attachment representation showed increased neural activity in the aMCC related to the monadic pictures. The findings also showed increased neural activity in the right STS as well as decreased neural activity in the right parahippocampal gyrus related to dyadic pictures that showed activations in the aMCC when confronted with “monadic” AAP scenes representing aloneness compared to healthy controls [21]. In both studies, the participants were instructed to tell stories to the AAP pictures while being scanned.

In a further study on BPD patients, Bernheim et al. [32] used an AAP attachment paradigm (see Buchheim et al., 2012 [33]) by presenting core sentences from their own AAP attachment narratives in the fMRI scanner. BPD patients showed increased fMRI-activation in brain areas associated with fear, pain, and hyperarousal than HC when presented with personalized attachment-relevant stimuli alone. Pictures with monadic attachment situations induced greater anterior medial cingulate cortex, anterior insula, amygdala, thalamus, and superior temporal gyrus activation in the patient group. The results point to increased fMRI-activation in areas processing emotional distress and painful experiences in BPD patients. In particular, the emotional cascade reflecting attachment distress was evoked by combining monadic pictures, representing abandonment and aloneness, with the patients’ personalized narrative material.

In a “Cyber-Ball” paradigm study focusing on social exclusion, BPD patients displayed an increased activity in dACC (≙aMCC), dorsolateral, as well as dorsomedial prefrontal cortex (dlPFC, dmPFC), precuneus, and anterior insula [34]. Another study showed hyperactivation in nucleus accumbens (NAcc) and amygdala during social pain experience [35]. 

Other previous imaging studies in BPD showed emotional hyperactivity represented by an increase of activation in the amygdala while watching emotional pictures as well as emotional faces [36,37]. Related to these findings, the presented pattern of results has been interpreted as BPDs being more likely to ascribe higher meaning to people’s intentions in ways of “hyper mentalization” [32,38]. 

Summarizing the findings so far, the following brain areas seem to build up an “attachment neural network” and play a role in specific attachment-relevant tasks: (1) mentalization-related processes and self-awareness: posterior superior temporal sulcus (pSTS), medial prefrontal cortex (mPFC), and temporal poles, (2) processes of perception/empathy of pain and fear: aMCC, amygdala, anterior insula, and (3) processes relevant for conflict monitoring, cognitive control, and reaction inhibition: aMCC, vPFC, and dlPFC [39]. 

The first fMRI study presenting longitudinal results during dialectical behavior therapy (DBT) in BPD patients reported on results of a small sample (n = 6). The authors used the international affective picture set (IAPS) to investigate response to differentially valanced and arousing pictures showing intervention results in a decreased fMRI-activation in aMCC and posterior cingulate cortex and left anterior insula. Further, for responders of treatment (n = 4), there was decreased activation in the amygdala and both sides of the hippocampi. Further, they reported a near-to-normal development in activation differences between HC and BPD group from baseline to post interventional imaging [40]. A meta-analysis additionally identified six fMRI-activation studies, investigating interventional effects of DBT linked to changes in emotional regulation capacity in BPD patients. Different paradigms for stimulating an emotional response by means of negative, positive, or neutral images including IAPS or a reappraisal task were associated with amygdala and anterior cingulate deactivation in BPD patients after DBT treatment [26]. 

### 1.4. Intervention Strategies

While BPD was considered difficult to treat for a long time, several studies have proven the opposite [6,41]. Evidence-based treatments for BPD [42,43] are well established today. Of which, Dialectical Behavioral Therapy (DBT) has published the most studies to demonstrate effectiveness [44]. DBT is a structured program of psychotherapy with a strong educational component designed to provide skills for managing intense emotions and negotiating social relationships. Originally developed to curb the self-destructive impulses of chronic suicidal patients, it is also the treatment of choice for borderline personality disorder, emotion dysregulation, and a growing array of psychiatric conditions. It consists of group instruction and individual therapy sessions, both conducted weekly, for six months to a year. The “dialectic” in DBT is an acknowledgment that real life is complex and health is not static, but an ongoing process hammered out through a continuous Socratic dialogue with the self and others. It is continually aimed at balancing opposing forces and investigating the truth of powerful negative emotions. 

Metaanalyses have demonstrated therapy success among DBT in terms of primary outcomes of BPD symptom severity, psychosocial functioning, self-harm behavior, and suicidality as well as secondary outcomes of specific BPD diagnostic criteria and depression [6,45]. However, studies with focus on neural changes of attachment representation are rare [33]. 

### 1.5. Study Aim and Hypotheses

Since a large body of research has reported a strong association between borderline pathology, adverse childhood experiences, and unresolved attachment, the present study examines neural changes of attachment in BPD patients during DBT treatment when confronted with their core issues of own attachment narratives for the first time [33,39]. Given the role of emotion regulation and impulsiveness in social cognitive impairment, interventions designed to improve emotion control, such as DBT, are assumed to be effective in improving social cognition and affecting regulation on a neural level. 

Our aim was to investigate the impact of one-year outpatient dialectical behavior therapy (DBT) on neuronal correlates of attachment representation in patients with BPD using a personalized fMRI task. For that, we used a standardized experimental setting, applying a fMRI-adapted AAP version including personalized core sentences from the participants’ own attachment stories, which is described in-depth in previous publications [18,33].

Along previous results (see Section 1.4), we expected an alignment of activation differences (near-to-normal development) between BPD and HC in attachment-associated brain areas e.g., amygdala and aMCC, particularly for monadic pictures paired with individual narrative sentences [39]. In addition, we tested associations between (1) fMRI-activation in regions-of-interest (ROI) and scores for global clinical staging of BPD (Borderline Personality Inventory-total score; BPI [46]) and the capacity for self-directedness (Temperament- and Character Inventory–self directedness scale; TCI_SD; [47]) at baseline and (2) the main treatment outcome parameters (changes in BPI, TCI_SD) and fMRI-activation in ROIs at baseline. 

## 2. Materials and Methods

### 2.1. Procedure and Study Design

The present study is part of a major functional magnetic resonance imaging study in borderline patients treated with outpatient DBT investigating changes in neuronal correlates of attachment representation in BPD by using a personalized functional magnet resonance imaging task. A much more detailed description of study participant characteristics, treatment concept, diagnostic assessments, as well as inclusion and exclusion criteria is given in a previous publication focusing exclusively on behavioral data [48]. The study was approved by the Ethical Commission of the University Medicine of Greifswald (BB 136/10). All participants gave informed consent to scientific procedures such as functional and structural MRI and data scoring before enrollment in the study.

BPD patients were tested before and after intervention with the clinical assessment battery as well as with structural and functional imaging (see below). HCs were only tested at baseline for clinical scores to compare with the BPD group. Imaging was performed twice for both subject groups in order to compare for changes between measurements without any interventions. Figure 1 depicts the overall study design.

### 2.2. Sample

The participants in the patient group before intervention were 26 exclusively right-handed female patients diagnosed with BPD aged 18 to 50 years, recruited from the Department of Psychiatry and Psychotherapy at the University of Greifswald and surrounding psychiatric hospitals. After the intervention period, we had 22 patients for clinical data assessment. For a second post intervention imaging, we had 18 BPD group participants. The study included patients with clinically relevant Borderline Personality Disorder symptoms using the Structured Clinical Interview for DSM-IV, axis II (SKID-II; cut-off ≥ 5 points) [49], the Borderline Personality Inventory (BPI) [46], and the Borderline Symptom List in its short version (BSL-23) [50]. Participants gave consent to a DBT-specific contract and, with that, participated in a 12-month outpatient DBT intervention. When consent was refused, participants were excluded from study participation. Further DBT exclusion criteria were being under legal guardianship, mental retardation (IQ < 70), florid psychotic symptoms, and powerful disorganization in cognition. Persons in whom contraindications for the fMRI examination were noted or who were pregnant during survey period were excluded from study participation. Due to comorbidity with other mental illnesses, medication with antidepressants seems to be unavoidable in BPD samples. Within our initial sample, 9 of 26 patients (34.6%) took selective serotonin reuptake inhibitors (SSRIs). Because of the strong altering effect on brain activity by antipsychotics, there were no participants on these drugs included in this study. Patients with medications received minimum and steady dosages over the whole study period.

A total of 26 healthy participants (HCs) were recruited via advertisements on university platforms and matched with patient groups in terms of age, sex, and education. Detailed baseline comparisons regarding the initial complete sample have been described before [39]. 

Inclusion criteria of the HCs were the willingness to participate in psychiatric assessments, attachment interviews, and an fMRI examination. Exclusion criteria were psychotherapeutic treatment and suffering from any current or former psychiatric disorder in HCs history. Furthermore, as in the BPD group, persons in whom contraindications for the fMRI examination were noted or who were pregnant during survey period were excluded from study participation. After the intervention period, there were 23 HCs for clinical data as well as imaging data assessment. A flow chart provides more details on exclusion and inclusion of participants (Figure 2). Moreover, we give an overview about the distribution of age, education, self-directedness, borderline symptom severity, and attachment classification at baseline for BPD and HC participants who were included in the pre-post imaging data analyses (Table 1). 

### 2.3. Measurements

#### 2.3.1. Clinical Instruments

For the investigation of axis I and axis II diagnoses (DSM-IV), trauma history, level of crystallized intelligence, borderline- and global symptom severity in the intervention group as well as the control group, and well-established questionnaire- and interview methods have already been described in Bernheim et al., 2019 [48] and in a cross-sectional paper [39]. In addition, comorbidity and medication can be found in [48]. Clinical scores used for longitudinal analyses here were the BPI and the TCI_SD. The Borderline Personality Inventory (BPI; Leichsenring, 1997) is a 53-item self-report instrument designed to measure the borderline personality organization according to Kernberg (1967) and includes an overall value and 4 scales: “experience of alienation and identity diffusion”, “fear of closeness”, “primitive defense mechanisms and object relations”, and “defective reality testing”. Sufficient reliability (Cronbach’s α = 0.68–0.91, rtt = 0.73–0.88) and construct validity have been confirmed (vgl. Bernheim et al., 2019). The Temperament and Character Inventory (TCI; [47,51] sub score of self-directedness (TCI_SD; 44 items) is used to assess personality based on the biosocial model, which divides personality into independent dimensions of temperament and character, with the sub-score of the character dimension we consider at the individual level to be self-direction. This includes aspects such as sense of responsibility, hopeful purpose, self-acceptance, self-actualization, and resourcefulness [47,52]. The standardization of the TCI was based on a sample of 509 adult persons. Sufficient reliability (α = 0.67–0.83), especially high for the subscale “self-directedness” (α = 0.82), have been confirmed. Content validity inter alia is demonstrated by the correlation between the TCI scores and scales of the “Gießen-Test”. Due to a high reliability as well as high predictive quality and change sensitivity in the context of psychotherapy [53], we decided to use the subscale self-directedness in the original true–false version in the current study in an exposed way. 

#### 2.3.2. DBT-Intervention

All participants in the BPD group of 12 months of out-patient DBT experienced individual psychotherapy (1 h per week), group psychotherapy (e.g., skills training; 2 h weekly), and telephone coaching as needed according to [41,54]. Further, patients on medications had regular consultations with a psychiatrist. The skills training included the proven modules of DBT, i.e., skills to improve mindfulness, distress tolerance, emotion regulation, interpersonal effectiveness, and self-esteem [55]. The team was composed of one psychodynamic-orientated consultant psychiatrist and three qualified behavioral therapists (total percentage of 75% women). In addition, of all the individuals involved in therapy, three had DBT basic training prior to the intervention and further and one was an accredited DBT therapist and trainer according to the guidelines of the German DBT umbrella organization (DDBT e.V.). Professional supervision for individual and group psychotherapy was offered regularly every 4 weeks by a DBT-accredited master supervisor. Supplemental supervision was also provided as needed during high conflict and critical situations [53]. 

#### 2.3.3. Functional and Structural Imaging

Scanning procedures have been described previously in our cross-sectional article [39]. A 3T Siemens Magnetom Verio (Erlangen, Germany) with a 32-channel head coil was used to obtain a T1 whole-head volume for structural mapping, T2*-weighted echoplanar images (EPIs) for functional mapping, and gradient echoes for unwarping the EPIs. The EPIs were typified by a repetition time (TR) of 2000 ms, an echo time (TE) of 23 ms, a flip angle (fa) of 90°, and a field of view (FOV) of 208 mm. Each of the volumes consisted of 33 slices (axial; AC-PC oriented with an additional rotation of 20° to reduce susceptibility artifacts in the frontobasal region) with a voxel size of 2 mm × 2 mm × 3 mm and spacing between slices of 1 mm. For every participant, 756 EPI volumes of the whole head were obtained. A total of 34 phase and magnitude images were taken in the identical FOV using a gradient echo sequence (GRE) of TR = 488 ms, TE(1) = 4.92 ms, TE(2) = 7.38 ms, and fa= 60° to compute a field map to correct for geometric distortions in EPI images. The T1-weighted three-dimensional image (MPRAGE) was applied as a high-resolution spatial structural image. The total number of sagittal anatomical images/slices was 176 (R = 1900 ms, TE = 2.52 ms, fa= 90°, voxel size 1 mm isotropic, matrix size = 256 mm × 256 mm).

#### 2.3.4. Functional Imaging Paradigm and Personalized Core Sentences

Subjects were presented with an fMRI-adjusted version of the AAP [33]. The AAP has been proven to measure consistent psychometric characteristics and shows a high test-retest reliability, a high reliability between judges and discriminant, as well as convergent validity for the assessment of attachment representation in adults [11,12,18,39] The AAP assesses attachment representations by analyzing individual responses to eight attachment-activating drawings. These illustrate adults or children being alone in a scenario of loneliness, disease, disconnection, death, and potential abuse, or in such scenery including a second person. The set of drawings contains one neutral picture (e.g., two children playing ball), three monadic pictures that show a child or an adult alone (representing abandonment), and four dyadic pictures portraying child–child/child–adult/adult–adult interaction scenarios (representing state of interpersonal distress). The images are accompanied by a neutral description of what can be seen in the drawing (e.g., “Child at Window—a child looks out a window” or “Bed—a child and woman sit facing each other at opposite ends of the child’s bed”). According to these drawings, participants are asked to tell a story about the drawings corresponding standardized interview questions. 

All participants (HCs and BPD-group) were presented with personalized AAP core sentences from their own narratives in the fMRI procedure and asked to rate them after scanning. For each picture according to each subject, three personalized sentences were taken from individual narratives following the procedure outlined by [33]. The selected phrases represent three core aspects of the individual narratives: First, a description of the event (e.g., “This is a lonely child, cut off from the world”); second, thoughts and feelings induced by the drawing (e.g., “Nobody plays with her, she feels desperate”); and third, the outcome of the story (e.g., “She is helpless without any hope.”) [39]. Each trial went as follows: Presentation: sentence–picture–fixation cross (see Figure 2). Trials have been individualized for each participant by matching the corresponding AAP picture to the core statements made in the personal narratives. In addition to these personalized trials, the same pictures were applied for the creation of both neutral and non-individualized trials which exclusively used phrases that contained only descriptive components about the environmental elements portrayed in the scene to match them with corresponding AAP images. Non-personalized trials were the same for all participants in total. Stimuli presentation of given visuals within fMRI adjusted AAP paradigm took place in a predefined order [56]. True to this approach for each of the two conditions, personalized and non-personalized, 6 trials were run, each consisting of a set comprising 7 continuous trials which followed the standardized sequence of “sentence-picture-10s fixation period”. Three sets included seven personalized phrase-drawing-combinations and three sets contained seven neutral phrase-drawing-combinations, identical for all subjects. Overall, we presented 84 trials, including 42 personalized and 42 non-personalized trials in an alternating manner, with a total running time of 25 min. There were no intermissions between the trials, as they were presented in one pass. Using a questionnaire, the personal meaning in terms of emotionality and autobiography of each personalized sentence per image was assessed by the participants immediately after imaging [33,48]. For the temporal order of the fMRI-paradigm, see Figure 3. 

#### 2.3.5. Preprocessing of Imaging Data

Data were analyzed using SPM12 (Wellcome Department of Cognitive Neuroscience) implemented in MATLAB (MathWorks, Inc., Natick, MA, USA). Unwarping of geometrically distorted EPIs was performed in the phase encoding direction using the FieldMap Toolbox. Each time-series was realigned to the first image of each session and resliced. EPIs were co-registered to the T1-weighted anatomical image, and T1-weighted images were segmented to localize grey and white matter and cerebrospinal fluid. This segmentation was the basis for spatial normalization to the Montreal Neurological Institute (MNI) template using the DARTEL approach of SPM. Here, a group template was calculated, refined iteratively by diffeomorphic registration steps, co-registered to the SPM MNI template, and used to spatially normalize the functional images while applying a Gaussian kernel smoothing filter (9 × 9 × 9 mm full-width at half maximum) to improve spatial alignment and increase the signal-to-noise-ratio. Six movement parameters estimated during the realignment procedure were introduced into the model as covariates to control for variance due to head displacements. A temporal high-pass filter (128 s) was applied to remove slow signal drifts. Individual statistical maps for main effects (personalized monadic pictures/personalized dyadic pictures) and contrasts (personalized monadic pictures minus personalized dyadic pictures; monadic pictures + personalized minus neutral sentences) were calculated using the general linear model. First-level contrast images of each subject were used for group statistics calculated as a random-effect analysis at the second level. 

For the definition of ROIs, we focused on BPD-associated brain areas mentioned in previous research according to processes of perception and empathy of pain and fear: the aMCC [29,30] and the amygdala [39].

Anatomical classification of fMRI-activation (MNI-space) was performed using optimized masks as ROIs for each hypothesis. For the mid-cingulate, we used the Neuromorphometrics brain atlas and for the amygdala the Anatomy Toolbox Version 1.7 [57]. Highest fMRI-activation within these masks were evaluated for each participant using in house scripting as described before [58]. These betas were then further evaluated using SPSS in a secondary statistical evaluation (see below). 

#### 2.3.6. Statistical Comparisons

Statistical analysis of assessment data was executed with SPSS (IBM coop.) version 21. Sample size calculation was performed for the clinical outcome with a clinical effect size of 0.6 and α = 0.05 based on meta-analyses investigating the global effectiveness of DBT [59] and, furthermore, based on a study investigating effects of DBT on the neural correlates of affective hyperarousal in BPD [40]. 

For between-group comparison of ratings, chi-square tests were applied; it should be noted at this point that non-parametric tests are required, since a normal distribution of the data cannot be assumed. 

First-level SPMs were calculated using SPM12 (version 7219). When testing differences between groups (assuming higher fMRI-activation in BPD-patients at baseline), we used one-sided *t*-tests corrected for three comparisons (ROIs; corresponds to a two-sided *p* of <0.032). When testing changes over time, we compared differences between groups for the post-intervention time point (assuming that baseline differences are no more relevant after intervention). Pearson correlations were used to evaluate associations of fMRI-activation with the main outcome scores (BPI and TCI_SL) during baseline and for changes in score over intervention.

## 3. Results

### 3.1. Sample Characteristics 

As described in detail [39], the complete sample of n = 26 BPD patients scored significantly higher than HCs regarding borderline- and global symptom severity as well as number of trauma and Posttraumatic Stress Disorder and exhibited a comorbidity of five additional axis I diagnoses and one additional axis II diagnoses (DSM-IV). The participants of the BPD group (n = 18) and the HC group (n = 23) considered here who were included in the pre–post imaging data analyses were comparable for age and intellectual performance/IQ. BPD patients scored significant higher borderline symptom severity (BPI, BSL-23), lower self-directedness (TCI_SD), and, measured by the AAP, revealed different attachment classification compared to the HCs (HCs: predominant insecure-dismissing attachment classification vs. BPD: predominant insecure-preoccupied attachment classification) (see Table 1). 

### 3.2. Clinical Changes during Intervention

Figure 4 plots the BPI and TCI_SD scores of the participants. Throughout intervention, there was a decrease of BPI total score from 27.23 ± 5.9 to 18.74 ± 10.43 (t(22) = 3.52; *p* = 0.002; Figure 4A). TCI_SD improved over intervention from 20.19 ± 9.40 to 30.04 ± 7.81 (t(22) = −5.18; *p* = 0.001; Figure 4B). 

### 3.3. fMRI Data Results

#### 3.3.1. Between Group Comparisons (Monadic Pictures with Personalized Sentences)

Before intervention, BPD patents showed increased amygdala activation left t(50) = 2.83; *p* = 0.014; beta average/SD: HCs: 0.36 ± 0.19; BPD: 0.58 ± 0.33) and aMCC activation (t(50) = 2.28; *p* = 0.028; beta average/SD: HCs: 0.96 ± 0.56; BPD: 1.33 ± 0.60). fMRI-differences between groups were absent after intervention (amygdala left: t(39) = 0.11; n.s.; aMCC: t(39) = 1.69; n.s.). 

#### 3.3.2. Association between BPD Group fMRI Images and Clinical BPD Scores

When testing associations between clinical scores (BPI; TCI_SD and fMRI-activation in preselected ROIs (aMCC, amygdala, STS) during baseline, we found a positive correlation for BPI with left amygdala activation (r(52) = 0.44; p_c_ = 0.01) and a negative correlation for TCI_SD with left amygdala activation (r(52) = −0.42; p_c_ = 0.02; Figure 5A). When testing which fMRI-activation at baseline was predictive for improvements in clinical scores (BPI, TCI_SD), we found a negative correlation for changes in BPI: for the aMCC: r(23) = −0.56; *p* = 0.006; higher fMRI-activation in aMCC initially led to fewer changes in clinical BPD total score (Figure 5B).

## 4. Discussion

### 4.1. Significance of the Study and Main Findings

There has been no study to date that has addressed the change in neuronal correlates of attachment representation during DBT. Therefore, this study is currently the first to investigate the functional representation of attachment-relevant stimuli using the Adult Attachment Projective Picture System (AAP; [11]) in an interventional design. This study specifically investigated changes in functional representation of attachment relevant stimuli of BPD patients in the course of DBT. After DBT, we found a near-to-normal representation in selected ROIs when confronting BPD patients with monadic pictures paired with personalized sentences from their own AAP interviews. That was relevant for the left amygdala and the aMCC, which showed differences between groups before intervention. When testing associations of clinical scores (BPI, TCI_SD) over all participants (BPD, HCs) with fMRI-activation in ROIs, we observed a positive association of BPI and a negative association of TCI_SD with amygdala activation at baseline. When testing associations of fMRI-activation at baseline with clinical improvement (BPI), we found a negative association of aMCC activation. Therefore, those who showed high aMCC activation and were confronted with monadic scenes paired with personalized sentences initially showed poorer BPI decline over DBT. In the following, we discuss these findings and emphasize the special importance of amygdala and aMCC activation for attachment representation in BPD. 

### 4.2. Intervention Effects and fMRI Findings

Over the last years, several studies reported intervention effects of DBT in BPD [45]. We have also previously published most of our treatment effects from the current study. We observed a reduction of BPI total score [48] and an increase of TCI_SD [53]. We selected BPI as it is frequently used in assessment of borderline personality disorder in adults. It is based on Kernberg’s concept of borderline personality organization [60] and contains four scales of alienation experiences and identity diffusion, primitive defense mechanisms and object relations, deficient reality testing, and fear of closeness [46]. In contrast, TCI_SD describes personality assessment based on the biosocial model, dividing personality into independent dimensions of temperament and character, where the subscore of the character dimension considered by us at the level of the individual is self-directedness and has been selected to be especially sensitive for monitoring the therapeutic intervention here [53]. This includes aspects such as sense of responsibility, hopeful purpose, self-acceptance, self-actualization, and resourcefulness [52,61].

### 4.3. fMRI-Effects in the aMCC

At baseline, we observed an increase of aMCC activation in response to presented monadic pictures accompanied by personalized sentences in BPD patients. In studies with healthy participants, aMCC was related to processing attachment relevant stimuli such as social exclusion and increased attachment anxiety [27,29,30,62]. Further, aMCC has been suggested to be part of the neural network that processes physical as well as social pain [28,63]. Previous studies in BPD patients revealed higher aMCC activation than controls for attachment-relevant stimuli related to social pain [34] Moreover, longitudinal fMRI-activation studies on interventional effects of DBT related to emotional regulation during IAPS presentation in BPD patients described decreased activation in the aMCC after intervention [26]. With the present longitudinal fMRI analysis, we observed a convergence of activation differences between the HC and BPD group after one year of outpatient DBT according to attachment relevant stimuli presented in the AAP. According to this, our results go along with reported fMRI results in a current meta-analysis [26]. In addition, those areas with highest reduction over intervention were those showing relevant association with clinical scores. When investigating predictive value of fMRI-activation for clinical outcome under DBT-intervention, we found that aMCC activation in response to the personalized monadic pictures of the AAP at baseline showed negative associations with gain in BPI. Therefore, BPD patients who initially showed high activation in the aMCC benefited less from intervention with regard to specific borderline symptomatology, as represented by the BPI total score. In line with Iskric and Barkley-Levenson [26], imaging prior to therapy initiation could make a significant contribution to an individualized therapy approach and the resulting higher individual benefit from DBT treatment. Although higher aMCC activation at study enrollment predicted less improvement for BPD patients, DBT led to improved clinical functioning and reduced aMCC activation regardless of severity at enrollment.

### 4.4. fMRI-Effects in the Amygdala

At baseline, we observed higher activation in BPD patients in the left amygdala. This goes along with reports from a previous study in BPD patients, which also reported increased activity of the amygdala compared to HC when confronted with attachment relevant stimuli related to social pain [35]. Several other studies also found evidence of a higher activation level of the amygdala in BPD-patients while (1) watching a set of standardized emotionally stimuli [64] and (2) watching emotional faces [37]. As a result, the increased amygdala activation in BPD is associated with an intensified emotional response, which is of a longer duration and occurs even in the presence of low-grade stress-associated stimuli. Increased amygdala activation was therefore linked to ascribing higher meaning to people’s intentions in ways of “hypermentalization” [36].

Longitudinal studies on interventional effects of DBT in BPD patients showed decreased activation in the amygdala after intervention when observing the IAPS [26]. With the present longitudinal data analysis on functional imaging results, we observed a convergence of activation differences between the HC and BPD group after one year of outpatient DBT according to attachment-relevant stimuli presented in AAP. This “near-to-normal” development was found for the left amygdala when comparing both groups in terms of monadic picture stimuli paired with personalized sentences from one’s own narrative. 

Schmitgen et al. [65] found that amygdala and parahippocampus activation during a cognitive reappraisal task after viewing negative pictures, severity measures of BPD psychopathology, and grey matter volume of the amygdala provided the best predictive power for identifying DBT treatment non-responders. In this regard, we found associations over all participants for the BPI and the TCI_SD with fMRI-activation in the left amygdala at baseline. An increased activity in the left amygdala was therefore associated with an increased level of borderline personality organization according to Kernberg [60]. In addition, an increased activity in the left amygdala was associated with a lower degree of self-directedness, which manifests itself in decreased self-acceptance, low capacity for constructive reflection and self-mastery, goal-setting, and achievement [52,61]. Once again, our results highlight the “supportive role” of a well-developed capacity for “self-direction” to manage intense emotions—in the context, it focuses on presenting personalized attachment stimuli in an fMRI environment. This led to the conclusion that the development of self-directedness should be an explicit goal in the context of DBT [53]. 

### 4.5. Limitations

Attachment related pictures used in our study [56] were grouped with individualized sentences, which might hamper group comparisons (less emotional relevant sentences for HCs). However, it was well suited for the longitudinal analyses and quite specifically addressed social experiences such as feeling lonely and being left alone. In contrast, other studies who applied the IAPS [40] had been associated with circumscribed modulation of fMRI-activation [66]. We argue that the target variable to be addressed by the AAP drawings is at a much higher level of organization: an attachment representation. These drawings elicit internal working models derived from early childhood attachment experiences with the closest attachment figures, which Bowlby [23] suggests elicits specific patterns in cognition, feelings, and behavior. Mehlum interpreted a “mechanism of change” due to DBT intervention in BPD [67] which was based on a definition from Kazdin [68]: the “…mechanism explains how the intervention translates into events that lead to the outcome”. Overall, the neural network associated with attachment can be modulated by DBT in association with the clinical outcome. 

Moreover, we used an ROI-based method for data analysis, selecting activation maxima in predefined anatomical masks. This approach has the advantage of selecting the strongest effect in areas probably affected by susceptibility artifacts accompanied by spatial distortions, which could hamper voxel-based approaches e.g., due to registration issues. However, it does not allow such a precise spatial localization of effects in comparison to voxel-based analysis approaches. Again, using activation maximum in ROIs is beneficial for small study sizes to observe even small effects but might lead to an overestimation of results. 

Moreover, we did not repeat the AAP interview after one year. Therefore, we could not support our results on functional representation changes of the attachment representation network in the BPD group on a narrative behavioral level. However, our previous results of the same cohort showed a significant reduction of the emotional valence when confronted with the personalized sentences of the own AAP interview in the AAP questionnaire, as well as a more secure attachment style, measured by the ASQ questionnaire, after one year of DBT intervention in the BPD group [48]. 

Additionally, in Bernheim et al. [39], we described two relevant limitations: At first, we had a wide range of patients with comorbidity of Posttraumatic Stress Disorder (PTSD: 39%); however, no differences in the fMRI activation patterns could be found between our BPD patients with or without PTSD. Secondly, nine of n = 26 BPD patients (34,6%) took Selective Serotonin Reuptake Inhibitors (SSRIs), which might have an additional effect on fMRI activation. The medication dosage was kept at minimum throughout the study and subgroup analyses revealed no differences in the fMRI activation patterns between BPD patients with or without SSRIs. 

Furthermore, we have to state that a larger sample size might have been able to detect smaller effects. Finally, for some scores, we did not repeat clinical scoring after one year in the HC group. There might have been some changes in scoring over time in that group too, although they did not undergo DBT. 

## 5. Conclusions

In accordance with previous studies [33,39], the present study again confirmed that monadic attachment stimuli from the AAP interview combined with personalized attachment material seem to be suitable to activate the attachment system on a neuronal level. Monadic pictures from the AAP represent the experience of abandonment and loss, which is strongly associated with the trauma history of patients with BPD. 

We may conclude that neuronal changes in the amygdala—as well as in the aMCC—towards a “near to normal” activation when confronted with monadic, personalized attachment material after one year of DBT might be interpreted as a sign of the patients’ increased capability to regulate intensive emotions in the context of abandonment and loss—and therefore, “social pain”. In this regard, one year of outpatient DBT including weekly individual therapy and skills training could help to stabilize the internal working model of attachment towards a more modulated attachment activation. 

Future studies with greater sample sizes should aim to replicate these results in the course of DBT. Moreover, with respect to the predictive value of imaging before treatment and its prediction for improvement in clinical scores, future research should address the question of how to adapt the current DBT approach for the treatment of BPD patients with elevated aMCC prior to therapy initiation to achieve an even better response to DBT. In addition, imaging prior to therapy initiation could contribute to an individualized therapy approach and result in even higher individual benefit from DBT treatment for BPD patients.

## Figures and Tables

**Figure 1 brainsci-13-01001-f001:**
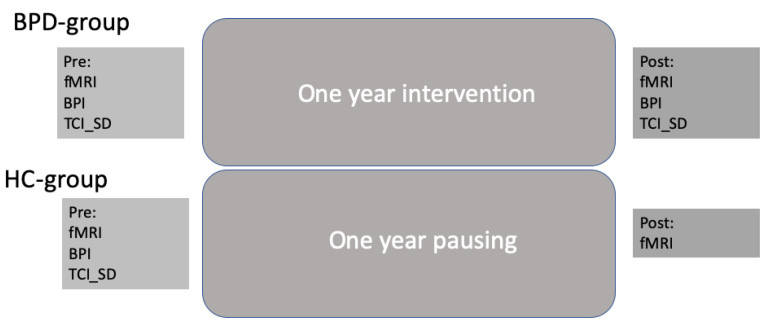
Overall study design.

**Figure 2 brainsci-13-01001-f002:**
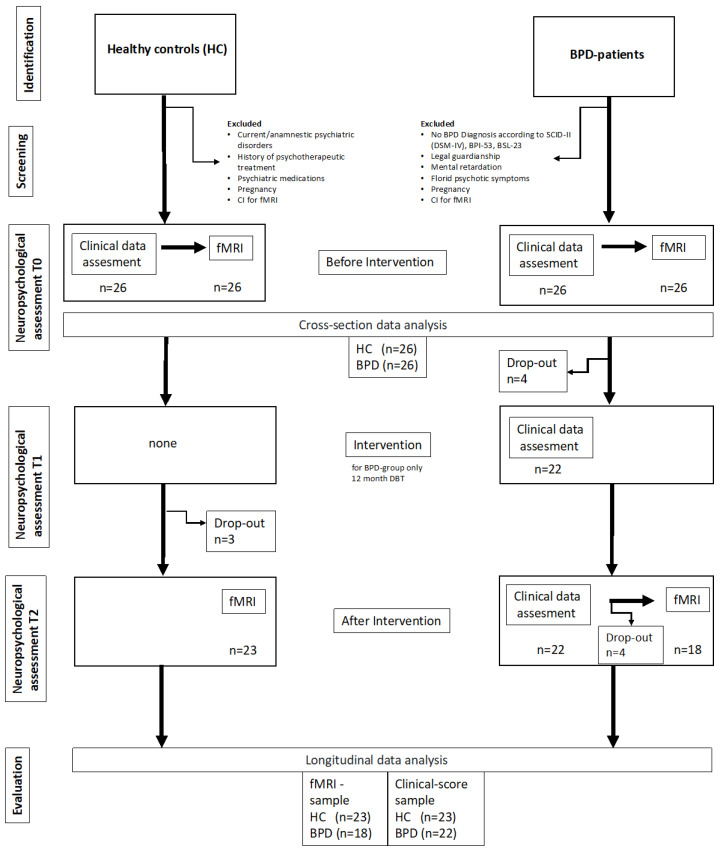
Flow chart of the study.

**Figure 3 brainsci-13-01001-f003:**
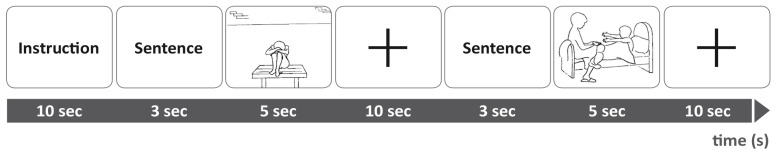
fMRI experimental procedure for a presentation of prior instructions, an emotional or neutral sentence for the monadic picture and a monadic picture. Then, a fixation cross indicated 10 s baseline and another emotional or neutral sentence is presented, followed by a dyadic picture [39].

**Figure 4 brainsci-13-01001-f004:**
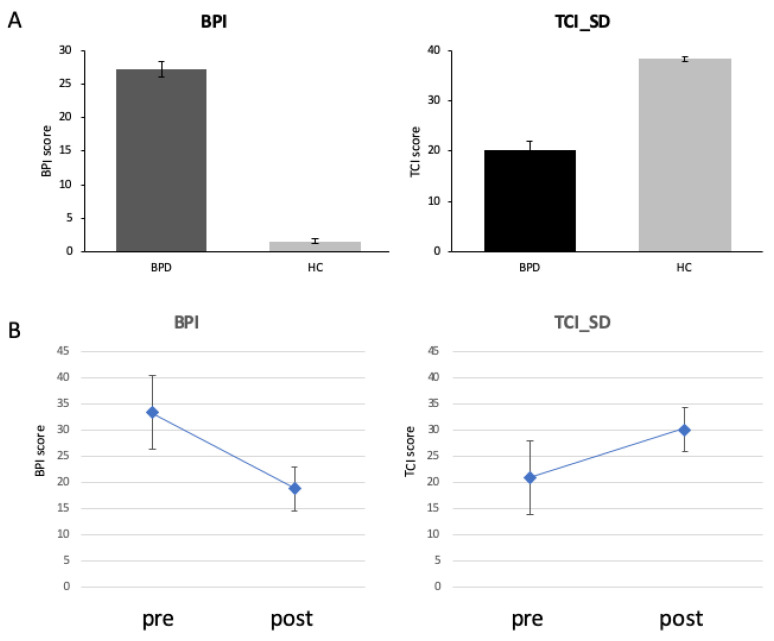
Clinical scores (BPI total score left and TCI_SD right) differed between groups at baseline (top, (**A**)) and after intervention (bottom, (**B**)).

**Figure 5 brainsci-13-01001-f005:**
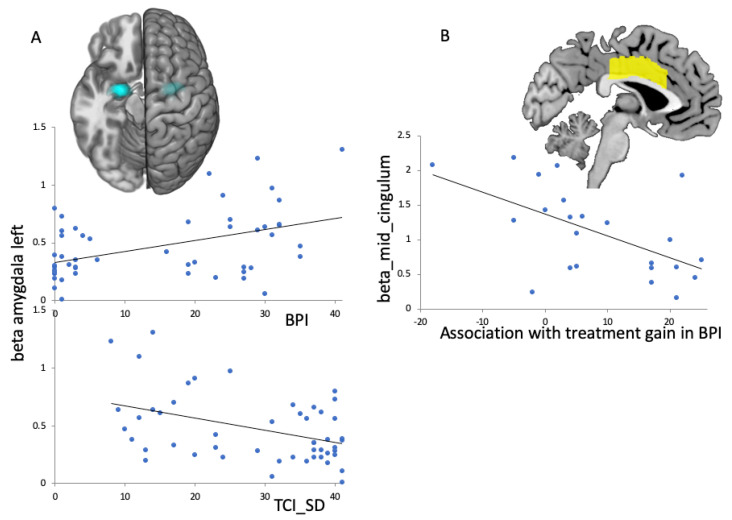
Linear regression plots of fMRI-activation in ROIs (amygdala and aMCC) and psychological scores. (**A**): Left amygdala activation at baseline (ROI indicated in blue) for all participants (HC and BPD) was associated with clinical scores at baseline; top: for the BPI total score there was a positive association; bottom: for the TCI_SD there was a negative association. (**B**): Anterior medial cingulate (aMCC) activation (ROI indicated in yellow) in BPD patients at baseline predicted changes in BPI total score over treatment.

**Table 1 brainsci-13-01001-t001:** Participant characteristics at baseline.

	Healthy Controls	BPD Patients	Statistics
Participants (n)	23	18	
Age in years	27.3 (6.83)	27.5 (7.97)	*t* = −0.78, *p* = 0.939
IQ (MWT-B)	108.4 (9.29)	111.7 (14.88)	*t* = −0.82, *p* = 0.419
TCI_SD	38.43 (2.66)	19.67 (9.81)	*t* = 7.893, *p* = 0.000
AAP_CLASS	2.70 (1.063)	1.83 (0.924)	*t* = 2.775, *p* = 0.008
BSL_total_score	2.0 (2.8)	44.2 (21.82)	*t* = −8.147, *p* = 0.000
BPI_sum	1.55 (1.82)	27.78 (5.86)	*t* = −18.295, *p* = 0.000

Mean values are presented with standard deviation in brackets. MWT-B= number of correctly recognized words, AAP_CLASS = AAP classification: 0 = cannot classify, 1 = unresolved, 2 = insecure-preoccupied, 3 = insecure-dismissing, 4 = secure. IIP-C = Interpersonal problems, TCI_SD = Temperament and Character Inventory_self-directedness, BSL_total_score= Borderline Symptom List, BPI_total_score= Borderline Personality Inventory.

## Data Availability

The data for this study can be handed out when needed. Please contact the corresponding author.

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
