# Peer review of "One Year of Outpatient Dialectical Behavioral Therapy and Its Impact on Neuronal Correlates of Attachment Representation in Patients with Borderline Personality Disorder Using a Personalized fMRI Task"

_brainsci, 2023, doi:10.3390/brainsci13071001_

Round 1

Reviewer 1 Report

Thank you for the opportunity to review this work. the manuscript covers a range of research on attachment, BPD, DBT effects, the roles of aMCC and amygdala, longitudinal and predictive neuroimaging studies. However, the manuscript still has the following problems worthy of attention, through the improvement of these problems can better improve the quality of the manuscript.

Abstract:

1. The background part requires clarification and more concise framing of the research gap and questions. Consider rephrasing to something like:

BPD is characterized by disturbances in attachment, but neuroimaging studies investigating attachment representations in BPD are lacking. No study has examined longitudinal neural changes associated with interventions targeting these disturbances. We aimed to address this gap by...” 

2. The methods part would benefit from more details on the study design, participants, interventions, imaging procedures and data analysis approaches.

3. The conclusions part needs to more comprehensively highlight the significance, implications and future directions of this study.

4. Additional suggestions: Use active voice as much as possible; avoid very lengthy and complex sentences; make sure each part connects logically to convey a clear summary and flow.

Introduction:

1. The introduction part lacks a clear thesis statement to convey the main focus and objective of the study. A concise thesis statement should be added at the end of the introduction.

2. The characterization of BPD in the first paragraph is too concise. More details on the symptoms, diagnosis criteria, prevalence, impacts, and prognosis of BPD should be provided to help readers understand the disorder comprehensively. Relevant statistics and data can also be supplemented.

3. The second paragraph on attachment studies is descriptive but lacks coherence and logic flow. The links between different concepts such as attachment styles, traumatic fear indicators and unresolved attachment should be explained more clearly. How these concepts relate to BPD should also be elaborated.

4.  The third paragraph summarizes fMRI paradigms and findings on attachment studies but the summaries lack details and depth. More explanations should be provided on what the paradigms are measuring and what the findings suggest. The links between different findings should also be clarified. References for major paradigms and findings should be supplemented.

5. The fourth paragraph on intervention strategies is too concise. More details on the interventions especially DBT should be provided, including the components, mechanisms, treatment procedures and outcomes. Relevant references should also be added.

6. The fifth paragraph states the study aim and hypotheses but lacks sufficient explanatory details. The authors should explain why they focus on attachment representation and how DBT may impact its neural correlates. More details on the fMRI task and what is expected to change should be provided to help readers understand the rationale and significance of the study.

7. Additional suggestions: Add more transition words or sentences to enhance the coherence and flow of ideas in the introduction; check if all statements are supported by up-to-date academic references.  If there are, the introduction part can be more persuasive.

Materials and Methods

1. In the procedure part, more details on the study design, procedure and timeline should be provided to help readers understand how the study was conducted. A concise flow chart can also be added for illustration.

2. In the sample part, the inclusion and exclusion criteria for both patient group and control group lack details. Specific cut-off scores and threshold for relevant clinical instruments should be provided. The demographic information of participants such as age, gender ratio, education level, symptom severity, medication status etc. should be summarized in a table for easy reference and comparison between groups.

3. The descriptions of measurements and materials, especially the clinical instruments and interventions, lack depth and supporting evidence. More details on the components, procedures, reliability, validity and previous research of relevant instruments/interventions should be supplemented with citations. The links between these instruments/interventions should also be clarified.

4. The imaging part requires more clarification and citations on previous literature to support the selected imaging techniques, parameters and task design. Detailed explanations on how personalized and non-personalized trials differ and what they aim to measure are needed. A model illustration on the imaging task can also be helpful for readers.

5. In the data preprocessing part, more details and illustrations (e.g. flow chart) are needed to help readers understand how the imaging data were processed step by step. Explanations for selecting specific methods/tools at each step should also be provided.

6. The procedure for defining ROIs requires more clarification. A figure showing the location and size of ROIs over brain template can be helpful for readers.

7. The statistical analysis part lacks sufficient details. For behavioral data, the statistical methods for different types of variables (e.g. continuous, categorical) should be specified. For imaging data, more explanation is needed on the individual and group level analysis as well as the specific t-test used (one sample? paired?). Details on multiple comparison correction methods should also be provided. The threshold for significance requires clarification.

Results

1. In the sample characteristics part, the information provided is very concise. More details on participants’ demographic and clinical characteristics should be provided in paragraphs for better reference and understanding.

2. The imaging results part requires more clarification and details. The locations (left, right) and coordinates of activated brain regions should be specified. Details on the t-tests used (one-sample, paired or independent?) and correction methods for multiple comparisons need to be provided.

Discussion

1. The current discussion section lacks depth in elaborating and interpreting the results. Much more in-depth discussions are needed on 1) the degree, implications and relations between changes observed over intervention; 2) the associations found between imaging and behavioral measures, including their directions, strengths, possible reasons and implications; 3) between-group differences found in brain activations, including their locations, implications and relations to previous literature.

2. Several parts require clarification and more supporting evidence. For example, explanations and examples are needed to clarify “a near-to-normal representation in selected ROIs”; citations are lacking for statements like “aMCC had been related to processing attachment relevant stimuli” and “Previous studies in BPD patients revealed...”. Direct comparisons of current results with previous findings can strengthen the persuasiveness.

Limitations and Conclusion

1. The limitations part requires substantial expansion. Much more discussion is needed on major limitations of the current study, including:

- Small sample size: The small sample may limit the generalizability of results and statistical power. Discuss how future research with larger sample can address this.

- Short intervention period: Discuss whether the intervention period of 1 year is sufficient for observing clinically significant changes. Longer follow-ups may be needed.

- Differences in stimuli used for fMRI: Discuss how differences between stimuli (personalized sentences vs IAPS) may influence the activation observed in ROIs.

- ROI-based analysis approach: Discuss both the advantages and disadvantages of using ROI-based analysis over voxel-based analysis. Explain how future research may employ voxel-based analysis for more precise localization of effects.

- Lack of follow-up behavioral measures: Discuss how repeating behavioral measures like questionnaires after intervention can provide further support for the imaging findings.

- Potential changes in control group: Discuss the possibility of changes over time in control group and the need to measure them in future research.

2. The conclusion part requires more concise and compelling statements that effectively highlight the significance, implications and contributions of the current study.

1. Ensure logical flow and coherence across sections. Use transition sentences to effectively link different ideas within and between sections. Reorganize or restructure parts if needed for enhanced clarity and coherence.

2. Strengthen the framing of research questions and hypotheses. Clearly state them early in the introduction and reiterate them at the beginning of the results and discussion sections. Discuss how findings relate to each question and hypothesis.

3. Enhance depth of discussions. Interpret results in more depth by discussing the degree, implications and possible reasons for observed changes or differences. Relate findings to previous literature and state how the current study contributes. Discuss both consistency and inconsistencies with prior research.

4. The language is generally clear and coherent, but some sentences are lengthy, complex or repetitive. Simplifying and tightening sentence structure and reducing repetition can make the language concise yet compelling. For example, “We argue, that the target variable to be addressed by the AAP drawings is at a much higher level of organization, an attachment representation.” can be rephrased as “We argue that the AAP drawings target attachment representations, which reflect a higher level of psychological organization.”

5. Transitions between ideas or sections are lacking in some places. Using transitional phrases like “however”, “moreover”, “in summary” and “in conclusion” can create logical connections and enhance the flow and coherence of language. For example, add a transition phrase in “At baseline, we observed an increase of aMCC activation in response to monadic pictures presented with personalized sentences in BPD patients. In studies with healthy participants, aMCC had been related to processing attachment relevant stimuli like social exclusion and increased attachment anxiety [26,28,29,56].”

6. Some statements lack supporting references or citations. Providing citations, especially when describing prior literature or findings, can substantiate the persuasiveness and credibility of claims. For example, add citations to support “Over the last years, several studies reported intervention effects of DBT in BPD [41].”

7. Avoid repetition across sections. Some ideas, especially descriptions of results, are repeated across multiple sections. Condensing repetitive parts and referring readers to previous sections can make the writing concise yet complete.

8. Abbreviations need to be standardized.

Author Response

Reviewer Responses

Reviewer 1:

Thank you for the opportunity to review this work. the manuscript covers a range of research on attachment, BPD, DBT effects, the roles of aMCC and amygdala, longitudinal and predictive neuroimaging studies. However, the manuscript still has the following problems worthy of attention, through the improvement of these problems can better improve the quality of the manuscript.

Response: Thank you for your overall positive evaluation of our manuscript.

Abstract:

  1. The background part requires clarification and more concise framing of the research gap and questions. Consider rephrasing to something like:

“BPD is characterized by disturbances in attachment, but neuroimaging studies investigating attachment representations in BPD are lacking. No study has examined longitudinal neural changes associated with interventions targeting these disturbances. We aimed to address this gap by...” 

Response: We thank the reviewer for his/her suggestion which we gratefully inserted now:

 “BPD is characterized by affect dysregulation, interpersonal problems and disturbances in attachment, but neuroimaging studies investigating attachment representations in BPD are rare. No study has examined longitudinal neural changes associated with interventions targeting these impairments.”

  1. The methods part would benefit from more details on the study design, participants, interventions, imaging procedures and data analysis approaches.

Response: We now inserted that in the Methods/Abstract line 22 ff: “We aimed to address this gap by performing a longitudinal neuroimaging study on n = 26 patients with BPD treated with Dialectic Behavioral Therapy (DBT) and n = 26 matched healthy controls (HCs; post intervention point: n = 18 BPD and n = 23 HCs)."

  1. The conclusions part needs to more comprehensively highlight the significance, implications and future directions of this study.

Response: We now changed the Conclusions part of the Abstract accordingly.

  1. Additional suggestions: Use active voice as much as possible; avoid very lengthy and complex sentences; make sure each part connects logically to convey a clear summary and flow.

Response: Now the sentences are less lengthy.

Introduction:

  1. The introduction part lacks a clear thesis statement to convey the main focus and objective of the study. A concise thesis statement should be added at the end of the introduction.

Response: The aims and hypotheses provide a clear thesis statement on aims and corresponding data analyses.

  1. The characterization of BPD in the first paragraph is too concise. More details on the symptoms, diagnosis criteria, prevalence, impacts, and prognosis of BPD should be provided to help readers understand the disorder comprehensively. Relevant statistics and data can also be supplemented.

Response: Thank you, we added some more information on BDP in the section 1.1.

  1. The second paragraph on attachment studies is descriptive but lacks coherence and logic flow. The links between different concepts such as attachment styles, traumatic fear indicators and unresolved attachment should be explained more clearly. How these concepts relate to BPD should also be elaborated.

Response: Thank you, we have now explained the concepts more clearly and added the association to BPD pathology in section 1.2.

  1. The third paragraph summarizes fMRI paradigms and findings on attachment studies but the summaries lack details and depth. More explanations should be provided on what the paradigms are measuring and what the findings suggest. The links between different findings should also be clarified. References for major paradigms and findings should be supplemented.

Response: Thank you very much for this suggestion. We added relevant explanations to explain what paradigms were measuring in the different studies and what results indicated in 1.3.

  1. The fourth paragraph on intervention strategies is too concise. More details on the interventions especially DBT should be provided, including the components, mechanisms, treatment procedures and outcomes. Relevant references should also be added.

Response:  We fully agree with the reviewer and added more information about components, mechanisms, treatment procedures and outcomes in 1.4.

  1. The fifth paragraph states the study aim and hypotheses but lacks sufficient explanatory details. The authors should explain why they focus on attachment representation and how DBT may impact its neural correlates. More details on the fMRI task and what is expected to change should be provided to help readers understand the rationale and significance of the study.

Response:  Thank you very much for this comment, we added some details to help readers understand the rationale and significance of the study.

  1. Additional suggestions: Add more transition words or sentences to enhance the coherence and flow of ideas in the introduction; check if all statements are supported by up-to-date academic references.  If there are, the introduction part can be more persuasive.

Response: Thank you we updated passages and included more transition sentences in the Introduction.

Materials and Methods

  1. In the procedure part, more details on the study design, procedure and timeline should be provided to help readers understand how the study was conducted. A concise flow chart can also be added for illustration.

Response: The differentiation in Procedure and Study Design might have been not a great idea. This has now been corrected and Procedure and Study design is under the same heading.  A study design Figure had been inserted as Figure 1. A very detailed description of the study had been provided before already (Bernheim et al., 2022). We already added a flow chart as Supplementary Figure but changed that now for Figure 2.

  1. In the sample part, the inclusion and exclusion criteria for both patient group and control group lack details. Specific cut-off scores and threshold for relevant clinical instruments should be provided. The demographic information of participants such as age, gender ratio, education level, symptom severity, medication status etc. should be summarized in a table for easy reference and comparison between groups.

Response: Thank you, we added some more details in 2.2. by presenting a flow chart as Figure 2.

  1. The descriptions of measurements and materials, especially the clinical instruments and interventions, lack depth and supporting evidence. More details on the components, procedures, reliability, validity and previous research of relevant instruments/interventions should be supplemented with citations. The links between these instruments/interventions should also be clarified.

Response: We agree with the reviewer and added more details on the components, procedures, reliability, validity and previous research of relevant instruments/interventions in 2.3.1

  1. The imaging part requires more clarification and citations on previous literature to support the selected imaging techniques, parameters and task design. Detailed explanations on how personalized and non-personalized trials differ and what they aim to measure are needed. A model illustration on the imaging task can also be helpful for readers.

Response: We demonstrated an illustration model on the imaging task in 2.3.4 and Figure 3. More detailed information on the imaging task have been provided in Buchheim et al. 2012 and Bernheim et al., 2022.

  1. In the data preprocessing part, more details and illustrations (e.g. flow chart) are needed to help readers understand how the imaging data were processed step by step. Explanations for selecting specific methods/tools at each step should also be provided.

Response: Descriptions on Imaging evaluations have now been extended.

  1. The procedure for defining ROIs requires more clarification. A figure showing the location and size of ROIs over brain template can be helpful for readers.

Response: The atlas for each mask have been cited in the Methods already. ROIS have already been depicted in Figure 5.

  1. The statistical analysis part lacks sufficient details. For behavioral data, the statistical methods for different types of variables (e.g. continuous, categorical) should be specified. For imaging data, more explanation is needed on the individual and group level analysis as well as the specific t-test used (one sample? paired?). Details on multiple comparison correction methods should also be provided. The threshold for significance requires clarification.

Response: We now inserted these details in the Methods.

Results

  1. In the sample characteristics part, the information provided is very concise. More details on participants’ demographic and clinical characteristics should be provided in paragraphs for better reference and understanding.

Response: These data have been provided in Table 1.

  1. The imaging results part requires more clarification and details. The locations (left, right) and coordinates of activated brain regions should be specified. Details on the t-tests used (one-sample, paired or independent?) and correction methods for multiple comparisons need to be provided.

Response: Coordinates: There are no such data for ROI-based analyses as performed here. Betas as fMRI-activation magnitude have now been provided for average and SD on page 11, line 386.

Discussion

  1. The current discussion section lacks depth in elaborating and interpreting the results. Much more in-depth discussions are needed on 1) the degree, implications and relations between changes observed over intervention; 2) the associations found between imaging and behavioral measures, including their directions, strengths, possible reasons and implications; 3) between-group differences found in brain activations, including their locations, implications and relations to previous literature.

Response: Thank you, all these details have already been provided in the Discussion. Directions of associations have even been plotted in Figure 5.

  1. Several parts require clarification and more supporting evidence. For example, explanations and examples are needed to clarify “a near-to-normal representation in selected ROIs”; citations are lacking for statements like “aMCC had been related to processing attachment relevant stimuli” and “Previous studies in BPD patients revealed...”. Direct comparisons of current results with previous findings can strengthen the persuasiveness.

Response: “Near to normal representation” is a term used in clinical Neurology especially after interventions. If representation intensity in ROIs is different for a certain paradigm initially but not different to HCs after intervention a near to normal representation is interpreted.

Limitations and Conclusion

  1. The limitations part requires substantial expansion. Much more discussion is needed on major limitations of the current study, including:

- Small sample size: The small sample may limit the generalizability of results and statistical power. Discuss how future research with larger sample can address this.

Response: We now inserted on page 15: Larger samples sizes might detect smaller effects.

- Short intervention period: Discuss whether the intervention period of 1 year is sufficient for observing clinically significant changes. Longer follow-ups may be needed.

Response: Since clinical scores changed drastically (see also Bernheim et al. 2018; 2019) longer intervention periods are not necessary.

- Differences in stimuli used for fMRI: Discuss how differences between stimuli (personalized sentences vs IAPS) may influence the activation observed in ROIs.

Response: We already detected highest differences between groups for monadic pictures before (Bernheim et al., 2022) We therefore would not like to discuss different paradigms here.

- ROI-based analysis approach: Discuss both the advantages and disadvantages of using ROI-based analysis over voxel-based analysis. Explain how future research may employ voxel-based analysis for more precise localization of effects.

Response: That has already been discussed in our previous version on page 14: “Moreover, we used a ROI-based method for data analysis selecting activation maxima in predefined anatomical masks. This approach has the advantage of selecting the strongest effect in areas probably affected by susceptibility artifacts accompanied by spatial distortions which could hamper voxel-based approaches. However, it does not allow such a precise spatial localization of effects in comparison to voxel-based analysis approaches. Again, using activation maximum in ROIs is beneficial for small study siz-es to observe even small effects but might lead to an overestimation of results.”

- Lack of follow-up behavioral measures: Discuss how repeating behavioral measures like questionnaires after intervention can provide further support for the imaging findings.

Response: This does only refer to the HCs. Therefore that is not related to our patient group here. 

- Potential changes in control group: Discuss the possibility of changes over time in control group and the need to measure them in future research.

Response: We definitely think that this would expand our manuscript into the wrong direction.

  1. The conclusion part requires more concise and compelling statements that effectively highlight the significance, implications and contributions of the current study.

Response: Thank you we added some concluding thoughts to highlight the significance, implications and contributions of the current study.

Comments on the Quality of English Language

  1. Ensure logical flow and coherence across sections. Use transition sentences to effectively link different ideas within and between sections. Reorganize or restructure parts if needed for enhanced clarity and coherence.
  2. Strengthen the framing of research questions and hypotheses. Clearly state them early in the introduction and reiterate them at the beginning of the results and discussion sections. Discuss how findings relate to each question and hypothesis.
  3. Enhance depth of discussions. Interpret results in more depth by discussing the degree, implications and possible reasons for observed changes or differences. Relate findings to previous literature and state how the current study contributes. Discuss both consistency and inconsistencies with prior research.
  4. The language is generally clear and coherent, but some sentences are lengthy, complex or repetitive. Simplifying and tightening sentence structure and reducing repetition can make the language concise yet compelling. For example, “We argue, that the target variable to be addressed by the AAP drawings is at a much higher level of organization, an attachment representation.” can be rephrased as “We argue that the AAP drawings target attachment representations, which reflect a higher level of psychological organization.”
  5. Transitions between ideas or sections are lacking in some places. Using transitional phrases like “however”, “moreover”, “in summary” and “in conclusion” can create logical connections and enhance the flow and coherence of language. For example, add a transition phrase in “At baseline, we observed an increase of aMCC activation in response to monadic pictures presented with personalized sentences in BPD patients. In studies with healthy participants, aMCC had been related to processing attachment relevant stimuli like social exclusion and increased attachment anxiety [26,28,29,56].”
  6. Some statements lack supporting references or citations. Providing citations, especially when describing prior literature or findings, can substantiate the persuasiveness and credibility of claims. For example, add citations to support “Over the last years, several studies reported intervention effects of DBT in BPD [41].”
  7. Avoid repetition across sections. Some ideas, especially descriptions of results, are repeated across multiple sections. Condensing repetitive parts and referring readers to previous sections can make the writing concise yet complete.
  8. Abbreviations need to be standardized.

Auch hier-scheint alles eher default-Info zu sein als spezifisch zu unserem Manuskript passend…

Eine einheitliche nette Antwort wär wohl zielführend wie zB: „

Response: We would like to thank the reviewer for his/her detailed comments. We additionally asked a native English speaker to correct our manuscript for problems in English and indicated the changes in the new manuscript. Abbreviations have been standardized now (schaust Du bitte noch mal durch Ariane?).

Reviewer 2 Report

The authors are to be congratulated on an important contribution in which they have documented changes in clinical and regional brain networks associated with attachment in patients with BPD treated for one year with DBT.  The sample included 26 BPD patients were women aged 18 to 50 years diagnosed with the Borderline Personality Inventory (BPI) and 26 healthy controls. Differences in clinical and neuronal features were documented between patients and controls and in response to treatment after 1 year. After 1 year of treatment there were 22 patients and 18 of these completed a second post intervention fMRI.

Clinically subjects were assessed with BPI, IQ testing, and the Self-directedness scale of the Temperament and Character Inventory (TCI).  BPI and TCI_SD allowed measurement of clinical change.  Attachment was assessed by the Adult Attachment Projective Picture System. DBT was provided by trained DBT therapists who were supervised every 4 weeks by a DBT-accredited master supervisor with additional supervision as needed during critical situations. fMRI used an fMRI-adjusted version of the AAPP in which subjects are shown 8 attachment-activating drawings in which people are alone in a scenario of loneliness, disease, disconnection, death and potential abuse, or in such scenery including a second person.  Each set includes a neutral picture, 3 monadic pictures with a person alone, and four dyadic pictures representing interpersonal distress.  These pictures are paired with a personalized statement of the subject from their clinical assessment at baseline. These personalized statements facilitate assessment of change in reactivity after treatment. The baseline clinical and fMRI findings were predictive of response to treatment and the changes in BPI (severity of borderline features)and TCI_SD (measure of being resourceful, purposeful, self-accepting, responsible, and self-actualized) were correlated with activity in regional brain networks involving self-awareness (posterior superior temporal sulcus, medial prefrontal cortex, and temporal poles), perception of pain and fear (anterior midcingulate cortex, amygdala, anterior insula), and conflict monitoring, cognitive control and reaction inhibition (anterior midcingulate cortex, ventral prefrontal cortex, and dorsolateral prefrontal cortex).

 Greater anterior cingulate activation at baseline predicted poorer treatment response, but was reduced with improvement in BPI following treatment. Self-directedness improved with DBT treatment, and this was associated with reduced amagdalar activation (Figure 4).

 I have only a few minor comments for improving the paper.  On line 52, what is PTBS?  The acronym is never written out.  It could be post-traumatic brain syndrome (PTBS) but I wonder if it refers to PTSD (post-traumatic stress disorder) which is common in BPD patients and their parents.  I did not find an answer to my question in the two cited references (7,8) by Patricia Cohen and Cathy Widom, who emphasize socioeconomic factors, parental maladjustment, and broad measures of abuse and neglect, so I am also uncertain what is meant by "difficult methods of education" on line 51.  The references do document impaired parental behavior and mental disorders, physical abuse, and neglect, as is well known.

Otherwise, the content of all sections of the article is excellent and the conclusions are well justified and well-discussed.  My main recommendation is that the article could be improved by being edited for use of the English language.  There are many statements, even in the abstract, that are understandable but not good English. For example, in the abstract, line 28, “When investigation associations between scores and function activation we found…” should be “When investigating…functional activation, we found …”. On line 33, “Over successful DBT treatment these” should be “After…treatment, these”.  Likewise, page 4, line 166, “the being under…” should be “being under…” with “the” deleted. Page 11, 431, “did less well profit” should be “benefitted less”.

     Most of the English issues are minor, but there is one sentence at line 434-437 that is an important statement and not clearly written.  It now reads “In this regard, BPD patients with extreme increase in aMCC at baseline, once more, could benefit from DBT strategies, which helps to regulate painful emotions in the context of social closeness and distance.”  I was unsure what this meant at first. Then I found that the facts are that high activation of aMCC predicts less improvement than low activation. Nevertheless, there is reduction in aMCC activation with DBT treatment, so even subjects with high aMCC activation can improve, just less than those with lower aMCC activation.  Perhaps this could be more clearly written to say something like “Although higher aMCC activation at intake predicted less improvement, DBT led to improved clinical functioning and reduced aMCC activation regardless of severity at intake.”

     As the developer of the TCI, I have some comments on the use of the single scale of Self-directedness.  The use here is appropriate, informative, and well-justified by prior work by the authors and others.  It should be made clear that it appears the authors are using the original true-false version of the TCI based on the manual citation and the scores for healthy controls and BPD subjects in Table 1.

 I would also encourage the authors to consider use of a short form of the full TCI-R in future work or at least both the SD and Cooperativeness scales.  With use of a 5-point Likert Scale in the TCI-R, both temperament and character measures can be obtained reliably with 140 items. This provides temperament measures relevant to pain and fear reactivity (e.g., Harm Avoidance), impulsivity (e.g., Novelty Seeking), social approval seeking (e.g., Reward Dependence), and Persistence (resistance to being discouraged and upset).  The character measures (besides self-directedness) provide direct measures of interpersonal cooperation (tolerant, helpful, empathic, fairness, and compassion), and self-transcendence, which is an important aspect of self-awareness and mentalization.  All of these are relevant to the neuronal processes being measured, so the full test would allow a much deeper analysis and understanding of the process of psychotherapeutic change along with a language for describing personality structure that is useful in any person-centered and experiential psychotherapy, as well as providing access to the extensive information about the neurobiology of the TCI from other experimental and neuroimaging work. Permission to use the short TCI-R in German can be obtained from the non-profit Anthropedia Foundation at https://anthropedia.org.  However, the current work is already highly informative with only the TCI_SD scale, so I mention this only as a possible suggestion to improve and deepen this already excellent line of investigation.

See preceding comments recommending improved English clarity and grammar

Author Response

Reviewer 2:

The authors are to be congratulated on an important contribution in which they have documented changes in clinical and regional brain networks associated with attachment in patients with BPD treated for one year with DBT.  The sample included 26 BPD patients were women aged 18 to 50 years diagnosed with the Borderline Personality Inventory (BPI) and 26 healthy controls. Differences in clinical and neuronal features were documented between patients and controls and in response to treatment after 1 year. After 1 year of treatment there were 22 patients and 18 of these completed a second post intervention fMRI.

Clinically subjects were assessed with BPI, IQ testing, and the Self-directedness scale of the Temperament and Character Inventory (TCI).  BPI and TCI_SD allowed measurement of clinical change.  Attachment was assessed by the Adult Attachment Projective Picture System. DBT was provided by trained DBT therapists who were supervised every 4 weeks by a DBT-accredited master supervisor with additional supervision as needed during critical situations. fMRI used an fMRI-adjusted version of the AAPP in which subjects are shown 8 attachment-activating drawings in which people are alone in a scenario of loneliness, disease, disconnection, death and potential abuse, or in such scenery including a second person.  Each set includes a neutral picture, 3 monadic pictures with a person alone, and four dyadic pictures representing interpersonal distress.  These pictures are paired with a personalized statement of the subject from their clinical assessment at baseline. These personalized statements facilitate assessment of change in reactivity after treatment. The baseline clinical and fMRI findings were predictive of response to treatment and the changes in BPI (severity of borderline features)and TCI_SD (measure of being resourceful, purposeful, self-accepting, responsible, and self-actualized) were correlated with activity in regional brain networks involving self-awareness (posterior superior temporal sulcus, medial prefrontal cortex, and temporal poles), perception of pain and fear (anterior midcingulate cortex, amygdala, anterior insula), and conflict monitoring, cognitive control and reaction inhibition (anterior midcingulate cortex, ventral prefrontal cortex, and dorsolateral prefrontal cortex).

 Greater anterior cingulate activation at baseline predicted poorer treatment response, but was reduced with improvement in BPI following treatment. Self-directedness improved with DBT treatment, and this was associated with reduced amagdalar activation (Figure 4).

 I have only a few minor comments for improving the paper.  On line 52, what is PTBS?  The acronym is never written out.  It could be post-traumatic brain syndrome (PTBS) but I wonder if it refers to PTSD (post-traumatic stress disorder) which is common in BPD patients and their parents.  I did not find an answer to my question in the two cited references (7,8) by Patricia Cohen and Cathy Widom, who emphasize socioeconomic factors, parental maladjustment, and broad measures of abuse and neglect, so I am also uncertain what is meant by "difficult methods of education" on line 51.  The references do document impaired parental behavior and mental disorders, physical abuse, and neglect, as is well known.

Response: Thank you for your overall positive evaluation of our manuscript. Thank you for detecting that mistake- that is post traumatic stress disorder in a German abbreviation which we now eliminated.

Otherwise, the content of all sections of the article is excellent and the conclusions are well justified and well-discussed.  My main recommendation is that the article could be improved by being edited for use of the English language.  There are many statements, even in the abstract, that are understandable but not good English. For example, in the abstract, line 28, “When investigation associations between scores and function activation we found…” should be “When investigating…functional activation, we found …”. On line 33, “Over successful DBT treatment these” should be “After…treatment, these”.  Likewise, page 4, line 166, “the being under…” should be “being under…” with “the” deleted. Page 11, 431, “did less well profit” should be “benefitted less”.

 Response: Thank you for identifying these mistakes which we now corrected.

     Most of the English issues are minor, but there is one sentence at line 434-437 that is an important statement and not clearly written.  It now reads “In this regard, BPD patients with extreme increase in aMCC at baseline, once more, could benefit from DBT strategies, which helps to regulate painful emotions in the context of social closeness and distance.”  I was unsure what this meant at first. Then I found that the facts are that high activation of aMCC predicts less improvement than low activation. Nevertheless, there is reduction in aMCC activation with DBT treatment, so even subjects with high aMCC activation can improve, just less than those with lower aMCC activation.  Perhaps this could be more clearly written to say something like “Although higher aMCC activation at intake predicted less improvement, DBT led to improved clinical functioning and reduced aMCC activation regardless of severity at intake.”

Response: Thank you again for identifying this problem. We gratefully followed your advice and changed the sentence criticized accordingly.     

     As the developer of the TCI, I have some comments on the use of the single scale of Self-directedness.  The use here is appropriate, informative, and well-justified by prior work by the authors and others.  It should be made clear that it appears the authors are using the original true-false version of the TCI based on the manual citation and the scores for healthy controls and BPD subjects in Table 1.

Response: Thank you for your helpful comment. We now clarified that we used the original true-false version of the TCI (german version) based on the manual citation.

 I would also encourage the authors to consider use of a short form of the full TCI-R in future work or at least both the SD and Cooperativeness scales.  With use of a 5-point Likert Scale in the TCI-R, both temperament and character measures can be obtained reliably with 140 items. This provides temperament measures relevant to pain and fear reactivity (e.g., Harm Avoidance), impulsivity (e.g., Novelty Seeking), social approval seeking (e.g., Reward Dependence), and Persistence (resistance to being discouraged and upset).  The character measures (besides self-directedness) provide direct measures of interpersonal cooperation (tolerant, helpful, empathic, fairness, and compassion), and self-transcendence, which is an important aspect of self-awareness and mentalization.  All of these are relevant to the neuronal processes being measured, so the full test would allow a much deeper analysis and understanding of the process of psychotherapeutic change along with a language for describing personality structure that is useful in any person-centered and experiential psychotherapy, as well as providing access to the extensive information about the neurobiology of the TCI from other experimental and neuroimaging work. Permission to use the short TCI-R in German can be obtained from the non-profit Anthropedia Foundation at https://anthropedia.org.  However, the current work is already highly informative with only the TCI_SD scale, so I mention this only as a possible suggestion to improve and deepen this already excellent line of investigation.

Response: Thank you for this important advice which we will be happy to follow in our future research.

Reviewer 3 Report

This paper examines changes in regional brain activity in two regions of interest that are hypothesized to be involved in attachment-related phenomena following completion of dialectical behavior therapy (DBT) in a sample of patients with borderline personality disorder (BPD) followed up for one year.

As there is little work linking therapeutic change, basic psychological processes and their neural correlates in this patient population, the current study is a potentially important contribution to the literature.

There are certain aspects of the paper which would benefit from correction or clarification, and these are enumerated below:

1. In the introductory section, the relationships between childhood attachment, adult attachment, and BPD should be discussed in more depth. While a comprehensive review of the field would be beyond the scope of this paper, a mention of salient replicated findings would provide valuable context for the current research.

2. The statement that BPD was considered "untreatable" is an exaggeration. It was (and often is) considered difficult to treat and is the subject of significant stigma even among professionals / within healthcare systems; this could be stated with greater clarity.

3. Are there any prior studies linking specific neural regions or circuits with symptom dimensions or severity in BPD? These could be cited and discussed where appropriate.

4. What was the rationale for selecting self-directedness as an outcome measure in this study? Would other dimensions of character (particularly cooperativeness, which is directly linked to human social behavior) also change over the course of DBT and correlate with neural mechanisms subserving attachment and interpersonal relationships? 

5. Comorbidity with other disorders is the rule rather than the exception in BPD. What comorbidities (e.g., major depression, anxiety disorders, PTSD, substance use disorders, eating disorders) did the patients in the study sample have? How could these comorbidities have influenced the study findings (e.g., by acting as confounding factors)?

6. The authors have mentioned excluding patients on antipsychotics as they can "alter brain activity". However, SSRIs are also associated with changes in regional brain activity / metabolism. Was any attempt made to correct for this in the data analyses (e.g., by comparing patients who were receiving SSRIs with those who did not, or by examining correlations between SSRI use / dosage and changes in brain activity in the regions of interest?)

7. Was the sample size adequate (in terms of statistical power) to address the research questions? How was this checked?

8. The regions of interest selected by the authors are certainly associated with attachment, but they are also associated with a wide range of higher-order brain processes which could be activated by visual or verbal stimuli related to attachment. How could the authors be certain that the changes observed were directly (as opposed to indirectly or even epiphenomenally) linked to attachment? Was any baseline assessment of attachment carried out and correlated with baseline fMRI findings in the regions of interest?

9. The flow chart provided at the end of the paper is more suitable for a review article; it is not clear why it was included. If the authors wish to include a flow chart, it should be specific to the article's content and follow standard guidelines.

10. The Discussion requires a more in-depth coverage of several key issues (see points 1, 3-6 and 8 above) and a more balanced discussion of study limitations.

A certain degree of language editing is required to address several minor errors in sentence structure and word usage (e.g., "crystallized" instead of "crystalline" in line 201; "had" instead of "hat" in line 222

Author Response

Reviewer 3:

This paper examines changes in regional brain activity in two regions of interest that are hypothesized to be involved in attachment-related phenomena following completion of dialectical behavior therapy (DBT) in a sample of patients with borderline personality disorder (BPD) followed up for one year.

As there is little work linking therapeutic change, basic psychological processes and their neural correlates in this patient population, the current study is a potentially important contribution to the literature.

Response: Thank you for your overall positive evaluation of our manuscript.

There are certain aspects of the paper which would benefit from correction or clarification, and these are enumerated below:

  1. In the introductory section, the relationships between childhood attachment, adult attachment, and BPD should be discussed in more depth. While a comprehensive review of the field would be beyond the scope of this paper, a mention of salient replicated findings would provide valuable context for the current research.

Response: Thank you for this comment, we have added some more information on the  relationship between childhood attachment, adult attachment, and BPD

  1. The statement that BPD was considered "untreatable" is an exaggeration. It was (and often is) considered difficult to treat and is the subject of significant stigma even among professionals / within healthcare systems; this could be stated with greater clarity.

Response: “untreatable” was deleted and “difficult to treat” inserted instead.

  1. Are there any prior studies linking specific neural regions or circuits with symptom dimensions or severity in BPD? These could be cited and discussed where appropriate.

Response: Thank you for your comment. We now added results from a recently published relevant review in our introduction.

  1. What was the rationale for selecting self-directedness as an outcome measure in this study? Would other dimensions of character (particularly cooperativeness, which is directly linked to human social behavior) also change over the course of DBT and correlate with neural mechanisms subserving attachment and interpersonal relationships?

Response: Thank you for your suggestion. We now added the relevant information in „clinical instruments“.

  1. Comorbidity with other disorders is the rule rather than the exception in BPD. What comorbidities (e.g., major depression, anxiety disorders, PTSD, substance use disorders, eating disorders) did the patients in the study sample have? How could these comorbidities have influenced the study findings (e.g., by acting as confounding factors)?

Response: Thank you for this important comment. We integrated the relevant information in both sections: „results“ and „limitations“.

  1. The authors have mentioned excluding patients on antipsychotics as they can "alter brain activity". However, SSRIs are also associated with changes in regional brain activity / metabolism. Was any attempt made to correct for this in the data analyses (e.g., by comparing patients who were receiving SSRIs with those who did not, or by examining correlations between SSRI use / dosage and changes in brain activity in the regions of interest?)

Response: Thank you for your suggestion, we integrated the relevant information into the limitations.

  1. Was the sample size adequate (in terms of statistical power) to address the research questions? How was this checked?

Response: Thank you for your important question. We now inserted the following comment into the section 2.3.6. “Statistical comparisons”: “Sample size calculation was performed for the clinical outcome with a clinical effect size of 0.6 and α = 0.05 based on meta-analyses investigating the global effectiveness of DBT [60] and furthermore based on a study investigating effects of DBT on the neural correlates of affective hyperarousal in BPD [40].”

  1. The regions of interest selected by the authors are certainly associated with attachment, but they are also associated with a wide range of higher-order brain processes which could be activated by visual or verbal stimuli related to attachment. How could the authors be certain that the changes observed were directly (as opposed to indirectly or even epiphenomenally) linked to attachment? Was any baseline assessment of attachment carried out and correlated with baseline fMRI findings in the regions of interest?

Response: This is an important issue. In fact, we had only information from the Buchheim et al. studies on ROIs showing different activation effects in participants with BPD for the paradigm used here. Based on prior imaging studies using IAPS pictures we expected that highest effects would be in these two regions.

  1. The flow chart provided at the end of the paper is more suitable for a review article; it is not clear why it was included. If the authors wish to include a flow chart, it should be specific to the article's content and follow standard guidelines.

Response: Sorry for that mistake! The proper flow chart has now been inserted as Figure 2.

  1. The Discussion requires a more in-depth coverage of several key issues (see points 1, 3-6 and 8 above) and a more balanced discussion of study limitations.

Response: We fully agree and we have added more several key issues and a more balanced discussion of study limitations

Comments on the Quality of English Language

A certain degree of language editing is required to address several minor errors in sentence structure and word usage (e.g., "crystallized" instead of "crystalline" in line 201; "had" instead of "hat" in line 222

Response: corrected- thank you!

Round 2

Reviewer 1 Report

None

Reviewer 3 Report

The changes made in the revised manuscript are satisfactory in my opinion, and have addressed the concerns raised in my earlier report. I have no further major changes or corrections to suggest.